# Genomic and epidemiologic characteristics of SARS-CoV-2 persistent infections in California, January 2021 - July 2023

John M. Bell[1‡], Jesse Elder[1‡], Rahil Ryder[1‡], Emily A. Smith[2¤‡], Michelle Scribner[2], Sabrina Gilliam[1], Deva Borthwick[3], Megan Crumpler[4], Jacek Skarbinski[5], Christina Morales[1], Debra A. Wadford [1*]

1 Viral and Rickettsial Disease Laboratory, Center for Laboratory Sciences, California Department of Public Health (CDPH), Richmond, California, United States of America, 2 Theiagen Genomics, Highlands Ranch, Colorado, United States of America, 3 COVID Control Branch, Division of Communicable Disease Control, CDPH, Richmond, California, United States of America, 4 Orange County Public Health Laboratory, Santa Ana, California, United States of America, 5 Department of Infectious Diseases, Division of Research, Kaiser Permanente Northern California, Oakland, California, United States of America

‡ These authors contributed equally to this work and share first authorship.
¤ Current address: Center for Infectious Disease Research & Policy, Research and Innovation Office, University of Minnesota, Minneapolis, MN, USA
* Debra.Wadford@cdph.ca.gov

## Abstract

Novel SARS-CoV-2 variants demonstrating considerable intra-host evolution emerged throughout the pandemic. The persistent infections thought to give rise to these variants, however, have been difficult to identify at scale. This study sought to detect and characterize persistent infection cases in California using routine epidemiologic and genomic surveillance data. We identified 69 persistent infection cases with collection dates between January 2021 and July 2023 ranging from 21 to 400 days in duration, with an average of 44 days. Significant differences were identified in age distribution, sex, hospitalizations, and deaths between persistent infection cases and all sequenced California SARS-CoV-2 cases. Underlying health conditions were identified for the majority of cases with available medical records. In these cases, the Spike receptor binding domain was enriched for nonsynonymous mutations, and these mutations demonstrated convergent evolution indicative of immune evasion and were observed in previous persistent infections. We describe a 400-day B.1.429 infection that demonstrates substantial intra-host evolution, and a BA.5.11 persistent infection revealing apparent competition between two intra-host viral subpopulations. By establishing a framework for detecting persistent infections, this study lays the groundwork for other public health organizations to monitor and investigate highly divergent SARS-CoV-2 viruses.

**Data availability statement:** Sequence data generated by CDPH are routinely uploaded to both GISAID and NCBI (BioProject PRJNA750736). GISAID, SRA, and GenBank accession numbers for genomes used in this study are available in S1 Table. Note that some genome assemblies were rejected from GenBank and are therefore missing GenBank accession numbers, but assemblies can still be accessed on GISAID. We believe the reason for these rejections were false frameshifts. Correction of these frameshifts in the genome assemblies was not attempted, as it would have involved manual manipulation of the sequence data, which is prone to error and not an open-source solution.

**Funding:** CDPH/COVIDNet genomic surveillance work was funded in part by the US Centers for Disease Control and Prevention, Epidemiology and Laboratory Capacity (ELC) for Infectious Diseases ELC Enhancing Detection and ELC Expansion Cooperative Agreement 6 NU50CK000539. The funders had no role in study design, data collection and analysis, decision to publish, or preparation of the manuscript.

**Competing interests:** The authors have declared that no competing interests exist.

## Author summary

Genomic surveillance has been used to monitor the evolution and spread of SARS-CoV-2 variants throughout the pandemic. When a new variant emerges, it is often due to the accumulation of mutations during a persistent infection, i.e., in an individual who was unable to clear the virus after an infection. Using genomic and epidemiologic surveillance data, we identify 69 of these persistent infections in California and provide demographic and clinical characteristics of these infections compared to the broader population of SARS-CoV-2 infections. The identification of risk factors for persistent infections provides important insight into the epidemiology of SARS-CoV-2, while the identification of shared mutations between these infections enhances our understanding of SARS-CoV-2 evolution that may result in new variants. Ultimately, our work may help public health practitioners to better monitor persistent infections in the future, prior to the potential emergence and spread of novel variants into the community.

## Introduction

The emergence of new viral variants has been a hallmark of SARS-CoV-2 evolution through the course of the pandemic. While there are several hypotheses regarding potential sources of new, highly divergent variants such as Omicron, including cryptic transmission in isolated populations and reverse zoonosis, there is a general consensus within the scientific community that SARS-CoV-2 infections that persist in humans beyond the initial acute phase are likely contributors of these highly divergent sequences [1–3]. Here, we refer to these types of infections as persistent infections, although they are sometimes referred to as long-term infections [4,5] or chronic infections [6,7] in the literature.

Historically, evaluating the genomics of persistent infections has been primarily limited to prospective longitudinal case studies following individual patients [8–10], although some have studied prospective cohorts [11,12] and retrospective groups [13–15]. Some studies have identified accelerated viral evolution in persistent SARS-CoV-2 infections, which may explain the high degree of divergence seen in new variants [4,12,16–18]. Other studies have identified convergent evolution across different persistent infection cases, indicating a potential functional advantage conferred by specific mutations within a single host [10,19,20]. One challenge in studying persistent infections is the lack of uniform definition for "persistent infection". Minimum time frames for persistent infections have ranged upwards from as low as 14 days in the literature (cf. 14 days [21], 21 days [12], 26 days [22], 30 days [1]).Further, many academic and clinical studies often have access to data that is unavailable to public health departments, such as symptom onset and duration. There remains a need to further identify and characterize these types of infections at scale using population-level surveillance data to better understand public health implications and monitor trends over time.

The California Department of Public Health (CDPH) established the California SARS-CoV-2 Whole Genome Sequencing (WGS) Initiative, also known as "California COVIDNet", early in the COVID-19 pandemic to develop a robust SARS-CoV-2 genomic surveillance program [23]. This has resulted in the submission of several hundred thousand baseline surveillance SARS-CoV-2 sequences throughout California to public repositories. CDPH concurrently developed the Integrated Genomic Epidemiology Dataset (IGED), a database of epidemiologic data that links directly to SARS-CoV-2 sequence data. The IGED contains Electronic Lab Reports (ELR) submitted to the California Reportable Disease Information Exchange (CalREDIE) and is continually maintained as part of routine genomic surveillance.

Given the likely importance of persistent infections in shaping the duration and magnitude of the COVID-19 pandemic, we performed a retrospective investigation of California SARS-CoV-2 sequences and associated epidemiologic data. CDPH has been able to leverage California COVIDNet and IGED databases to analyze persistent infections at scale without the need for prospective studies, which are typically outside the scope of public health practice. To our knowledge, this is the first study to comprehensively detect and characterize persistent infection cases using a combination of genomic surveillance data, epidemiologic data, and clinical information. This study provides insights into intra-host evolutionary dynamics as well as potential epidemiologic factors associated with these cases.

## Results

### Retrospective analysis revealed 69 persistent infection cases in California

In the IGED, 987,973 sequenced SARS-CoV-2 infections with specimen collection dates through November 2023 were screened using the initial epidemiologic case criteria (Fig 1), of which 217 potential persistent infection cases were identified. The genomic data for these 217 cases were then used to classify them as persistent infections or reinfections, after which 69 were categorized as persistent infections (S1 Table). We defined persistent infections as a single, prolonged infection with specimen collection dates at least 21 days apart (cf. Methods), whereas a reinfection is characterized by separate infections from multiple strains of virus. The other 148 cases were excluded from the study; either they were clearly reinfections, or we were unable to distinguish between persistent infection and reinfection. The number of sequenced genomes for each persistent infection ranged from two to six, with 47 (68.1%) persistent infection cases having only two genomes.

### Persistent infections had an average duration of 44 days

The collection dates for the persistent infection cases ranged from January 2021 through July 2023 (Fig 2A). The number of days between the first and last reported specimen collection date for the 69 persistent infections ranged from 21 to 400, with an average of 43.8 days and median of 28 days. There were five persistent infection cases with a duration over 100 days, of which three were the Omicron variant, one Delta, and one Epsilon. The number of days reflects the duration between specimen collection dates, which may or may not reflect the actual duration of infection. The SARS-CoV-2 genomes of the 69 persistent infection cases belonged to six different WHO Variants (https://www.who.int/docs/default-source/coronaviruse/annex2_previous_vocs_and_definitions.pdf) and 17 different Nextclade clades [24] (Fig 2B). The number of consensus nucleotide changes between the first and last genome in each persistent infection case ranged from zero to 42, with an average of 4.9 nucleotide changes, a median of two nucleotide changes, and a mode of zero nucleotide changes. There was no evidence of onward community transmission from any of these persistent infection cases in California.

### Demographics differ between persistent infection cases and all sequenced SARS-CoV-2 cases

Of the 69 patients with persistent infections, 42 (60.9%) were male and 27 (39.1%) were female (Table 1). Comparatively, for all sequenced SARS-CoV-2 specimens in the IGED, 46.4% were male and 51.6% were female (sex was not reported

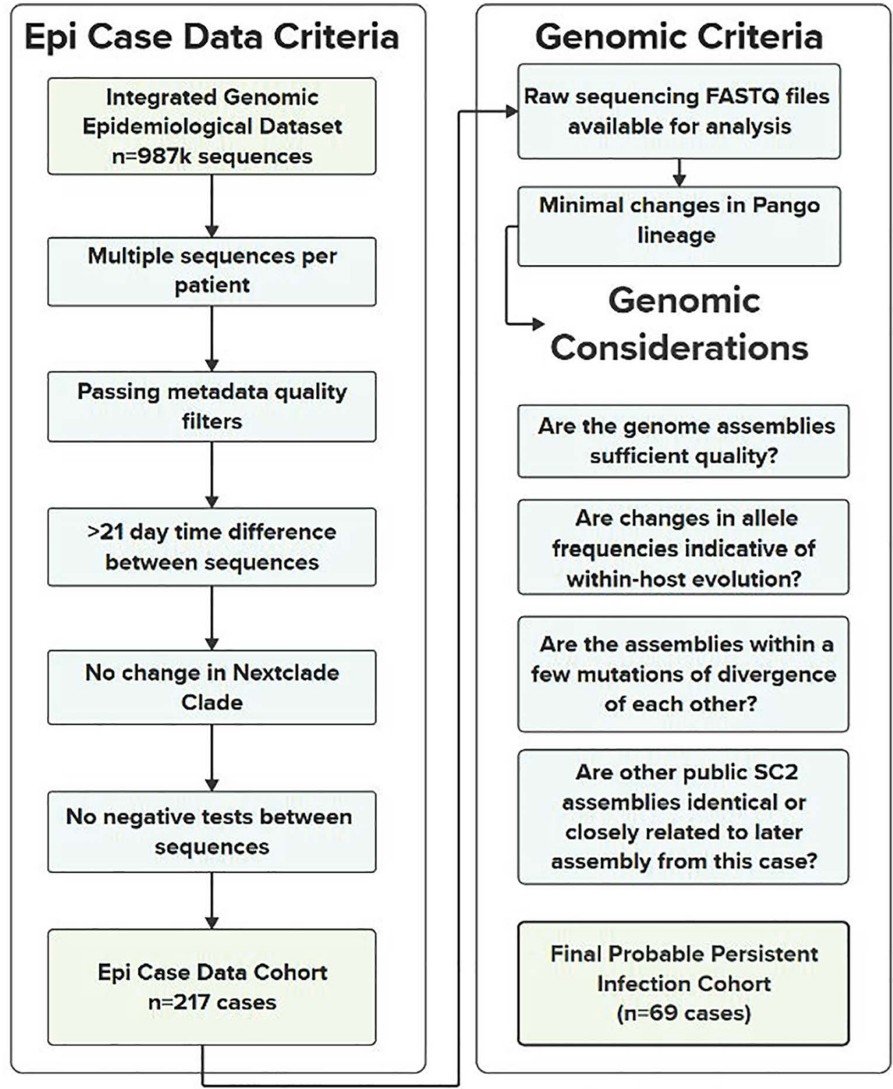

**Fig 1. Diagram of the epidemiologic and genomic case criteria used to obtain the final list of probable persistent infections.** More details on genomics considerations can be found in the S1 File.

for 2.0%). There was a significant difference in the numbers of males and females between these two groups ($\chi^2 = 4.5$, df = 1). The age groupings of patients with persistent infections included three (4.3%) under the age of 17, 20 (29.0%) aged 18–49, 14 (20.3%) aged 50–64, and 25 (36.2%) aged 65–79. For ages above 18, the age distribution between persistent infections and all sequenced specimens was significantly different ($\chi^2 = 81.4$, df = 3). The Association of Bay Area Health Officials region and the Southern California Health Officers region included 27 (39.1%) and 16 (23.2%) of the persistent infection cases, respectively. By contrast, of all California sequenced specimens, 62.0% were from the Southern California Health Officers region (S1 Fig). This difference may be due to data availability rather than true distribution of persistent infections in California. Of cases with known ethnicity (n = 59), 27 (45.8%) were White and 18 (30.5%) were Latino. From the California Immunization Registry, the COVID-19 vaccination status of 43 (62.3%) cases was available among the persistent infection cases. Of these, 23 had three to five vaccine doses while six had one or two doses.

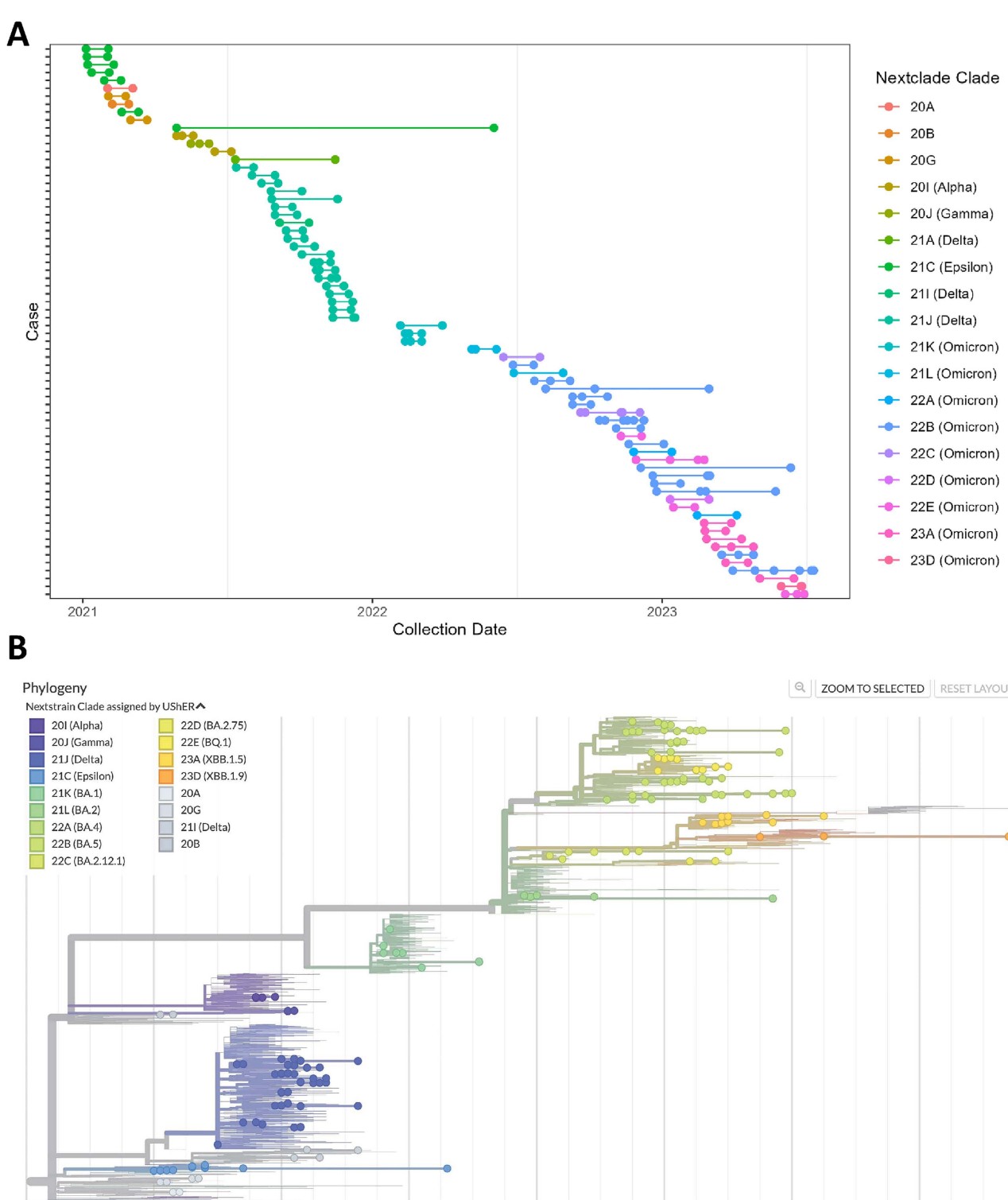

**Fig 2. Genomic diversity of 69 persistent SARS-CoV-2 infections from California. A** Dot plot showing collection dates of each sequence from persistent infection cases. Each row represents one case, each dot represents one sequence, and multiple horizontal dots indicate multiple sequences from a single case. **B** Phylogenetic tree of persistent infection sequences with global contextual sequences, colored according to Nextclade Clade assigned by UShER.

**Table 1. Demographics of all California cases with sequenced SARS-CoV-2 infections in the Integrated Genomic Epidemiology Dataset, all cases with putative persistent SARS-CoV-2 infections from this study, and a subset of cases with persistent infections with available medical records.**

| | | All Sequenced CA SARS-CoV-2 Infections | | All Persistent Infections | | χ² Results | Persistent Infections w/ Medical Record | |
|---|---|---|---|---|---|---|---|---|
| | | n | % | n | % | | n | % |
| | Total Patients | 812,563 | | 69 | | | 14 | |
| Sex | Female | 419,224 | 51.6% | 27 | 39.1% | χ²=4.5ᵃ df=1 | 6 | 42.9% |
| | Male | 377,074 | 46.4% | 42 | 60.9% | | 8 | 57.1% |
| | Unknown | 16,265 | 2.0% | | | | | |
| Age | ≤17 | 134,168 | 16.5% | 3 | 4.3% | | 0 | 0.0% |
| | 18-49 | 450,174 | 55.4% | 20 | 29.0% | χ²=81.4ᵃ df=3 | 1 | 7.1% |
| | 50-64 | 141,742 | 17.4% | 14 | 20.3% | | 2 | 14.3% |
| | 65-79 | 64,069 | 7.9% | 25 | 36.2% | | 8 | 57.1% |
| | ≥80 | 20,855 | 2.6% | 7 | 10.1% | | 3 | 21.4% |
| | Unknown | 1,555 | 0.2% | | | | | |
| Public Health Officer Region | ABAHO | 160,546 | 19.8% | 27 | 39.1% | | *Intentionally left blank* | |
| | Greater Sac | 52,813 | 6.5% | 11 | 15.9% | | | |
| | RANCHO | 12,361 | 1.5% | 3 | 4.3% | | | |
| | SJVC | 81,140 | 10.0% | 12 | 17.4% | | | |
| | Southern CA | 503,906 | 62.0% | 16 | 23.2% | | | |
| | Unknown | 1,797 | 0.2% | | | | | |
| Ethnicity | African American | 41,651 | 5.1% | 5 | 7.2% | | 1 | 7.1% |
| | American Indian | 2,735 | 0.3% | 1 | 1.4% | | 0 | 0.0% |
| | Asian | 77,752 | 9.6% | 4 | 5.8% | | 1 | 7.1% |
| | Latino | 284,545 | 35.0% | 18 | 26.1% | | 0 | 0.0% |
| | Native Hawaiian & other Pacific Islander | 5,461 | 0.7% | 3 | 4.3% | | 0 | 0.0% |
| | White | 181,495 | 22.3% | 27 | 39.1% | | 8 | 57.1% |
| | Other | 68,502 | 8.4% | 1 | 1.4% | | 0 | 0.0% |
| | Unknown | 150,422 | 18.5% | 10 | 14.5% | | 4 | 28.6% |
| Vaccine Dose Counts | 1-2 doses | 128,955 | 15.9% | 6 | 8.7% | | 1 | 7.1% |
| | 3-5 doses | 108,827 | 13.4% | 23 | 33.3% | | 12 | 85.7% |
| | Unknown | 359,442 | 44.2% | 26 | 37.7% | | 1 | 7.1% |
| | Vaccinated after | N/A | | 14 | 20.3% | | 0 | 0.0% |
| Hospitalized | Yes | 28,233 | 3.5% | 26 | 37.7% | χ²=230.4ᵃ df=1 | 9 | 64.3% |
| | No | 784,330 | 96.5% | 43 | 62.3% | | 5 | 35.7% |
| Died | Yes | 4,457 | 0.5% | 8 | 11.6% | χ²=134.5ᵃ df=1 | 2 | 14.3% |
| | No | 808,106 | 99.5% | 61 | 88.4% | | 12 | 85.7% |

For the Public Health Officer Region, ABAHO is the Association of Bay Area Health Officials, Greater Sac is the Greater Sacramento Area, RANCHO is the Rural Association of Northern California Health Officers, and SJVC is the San Joaquin Valley Consortium.

ᵃIndicates significant difference between Persistent Infection and All Sequenced CA SARS-CoV-2 datasets.

Fourteen cases were vaccinated after the persistent infection, and the vaccination status of the other 26 persistent infection cases in our study could not be determined. According to the COVID-19 Hospitalization Registry and COVID-19 Case Registry, 26 (37.7%) of the persistent infection cases were hospitalized and 8 (11.6%) died. Of those who died, three were infected with the Delta variant (ages 23–74), three were infected with Omicron BA.5 variant (ages 44–81), one was

infected with Omicron BA.2 variant (age 66), and one was infected with Omicron XBB.1.5 variant (age 90). Notably, there were significant differences in the number of hospitalizations ($\chi^2 = 230.4$, df = 1) and deaths ($\chi^2 = 134.5$, df = 1) between persistent infections and all sequenced SARS-CoV-2 cases in California (Table 1).

## Medical records indicate presence of comorbidities in subset of persistent infection cases

Of the 69 persistent infection cases, 14 (20.3%) had available medical records (Table 1) that were reviewed to determine their antiviral treatment regimen and evidence of a pre-existing condition indicative of a compromised immune system. The number of days between the first and last specimen collection date for these 14 persistent infection cases ranged from 26 to 189; however, one case reported an infection four months prior to sequencing and another case had a positive test one month before the first sequence. Eight (57.1%) cases were treated with remdesivir, three (21.4%) cases were treated with nirmatrelvir/ritonavir, and three (21.4%) cases did not receive antiviral treatment. Nine (64.3%) cases were hospitalized and two (14.3%) died, both of whom were male, had lymphoma (stage IV lymphoma, non-Hodgkin's lymphoma), were older than 65 years in age, and were infected with the variant Omicron BA.5 (BA.5.1.30 and BF.10 lineages, respectively).

Of the 14 cases with available medical records, nine (64.3%) cases were diagnosed with cancer, of which eight (88.9%) cases were lymphoma (non-Hodgkin's, mantle cell, diffuse large B-cell, and follicular) and one (11.1%) was hepatocellular carcinoma. In addition to cancer, some cases had other comorbidities such as chronic obstructive pulmonary disease (COPD), atrial fibrillation, chronic kidney disease, hypothyroidism, and epilepsy. Of those with cancer, five (55.6%) were confirmed to be undergoing treatments associated with immune suppression at the time of infection, including chemotherapy, Chimeric Antigen Receptor (CAR) T-cell therapy, and radiation therapy.

Of the five cases with available medical records and no evidence of cancer, four (80.0%) had underlying conditions that may have contributed to a compromised immune system. One case had chronic kidney disease and was taking immunosuppressive medications; the second case had rheumatoid arthritis and a history of Sjögren's syndrome; the third case had received a lung transplant and was taking immunosuppressive medication; and the fourth case had a renal transplant and was taking immunosuppressive medication. One case did not have enough information in the medical record to determine whether they may have had a compromised immune system during the persistent infection.

## Recurrent mutations demonstrate convergent evolution

Nonsynonymous consensus substitutions that arose during persistent infections were designated as having "accumulated" during the persistent infection. In total, 201 accumulated mutations were identified in specimens from the 69 persistent infection cases (S2 Table). Several of these consensus mutations occurred at the same genomic position in more than one individual, suggesting convergent evolution. There were 14 genomic positions where amino acid substitutions accumulated in multiple infections. Substitutions at S:444 accumulated in four separate infections, and substitutions at ORF1b:662 and S:484 accumulated in three separate infections each. The specific amino acid substitutions and the lineages in which they were found are shown in Table 2, which includes the public health significance associated with those changes. Substitutions at 11 other positions recurred in two infections (Fig 3A, S2 Table).

Recurring mutations at specific positions or in a specific region of the SARS-CoV-2 genome may be indicative of evolutionary pressures at those loci. The Spike receptor binding domain (RBD: nucleotide positions 22517–23185; Spike residues 319–541) exhibited mutations at several recurrent positions and certain regions showed more mutations than expected by chance. Bin 46 (nucleotide positions 22,500–23,000), which overlaps with the receptor binding domain (RBD), was significantly enriched (13 substitutions, adjusted p = 0.005) for nonsynonymous accumulated consensus mutations (Fig 3A). Mutations at residue 444 in the RBD of the Spike protein recurred most often, appearing in four infections – two observed as S:K444R (Pango lineages BA.4.6, BQ.1.25.1) and two as S:K444N (Pango lineages BA.2.12.1, BA.5.2.35) (Fig 3A, Table 2). We also observed three unique consensus changes at position 484, again in the Spike RBD: S:E484G (AY.2), S:E484K (AY.44), and S:E484T (BA.4.6). (Fig 3A, Table 2).

**Table 2. Amino acid positions with nonsynonymous consensus mutations observed in 3 or more cases.**

| Consensus Mutations | | | | |
|---|---|---|---|---|
| AA Position | AA Substitution | # of Infections | Observed Lineage | Description |
| ORF1b:662 | ORF1b:G662S | 2 | BA.5.1.30, BA.5.29 | Lineage-defining for Delta and XBB |
| | ORF1b:G662V | 1 | BA.5.2.1 | Unknown functional impact |
| S:444 | S:K444R | 2 | BA.2.12.1, BA.5.2.35 | [a] |
| | S:K444N | 2 | BA.4.6, BQ.1.25.1 | [a] |
| S:484 | S:E484G | 1 | AY.2 | [a] |
| | S:E484K | 1 | AY.44 | Present in JN.1, Beta, Gamma, other VOCs [ab] |
| | S:E484T | 1 | BA.4.6 | [ab] |

Lineages are shown for each unique amino acid substitution and known public health significance associated with those substitutions.

[a]Known immune escape mutation

[b]Known persistent infection or saltation lineage mutation

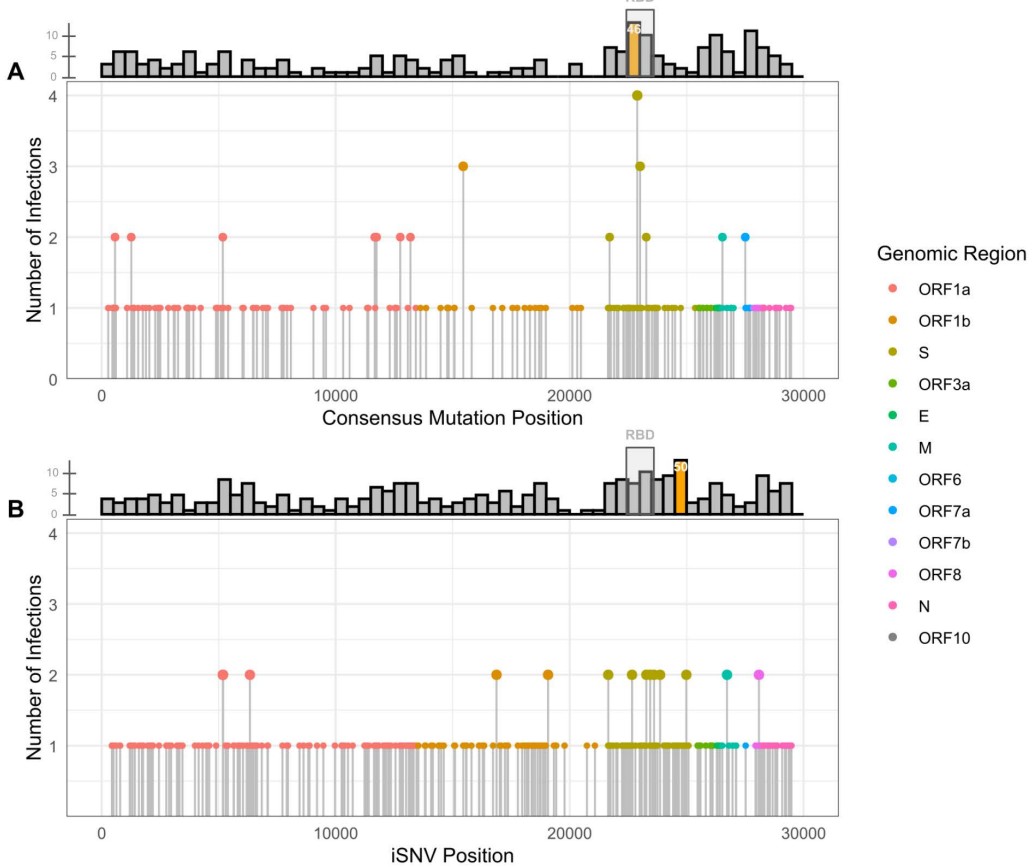

**Fig 3. Convergent mutations and nonsynonymous mutation enrichment in persistent SARS-CoV-2 infections. A.** Plot showing the number of infections in which amino acid substitutions accumulated in consensus genomes. **B.** Plot showing the number of infections in which amino acid substitutions appeared as intra-host single nucleotide variants (iSNVs). Filled circles are colored by genomic region. The location of each amino acid substitution is counted once per case, and any counts greater than one indicate a mutation occurred at the same position in more than one persistent infection. Histograms showing mutation counts in 500 nucleotide bins are shown in the top margin of each plot. Orange bars indicate genomic bins significantly enriched for mutations using a one-tailed binomial test after correction for multiple testing (Bonferroni adjusted p ≤ 0.05). The translucent grey rectangle in each histogram highlights bins overlapping with the Spike receptor binding domain (RBD: nucleotide positions 22517-23185; Spike residues 319-541). The bin numbers of significantly enriched bins 46 (nucleotide positions 22,500-23,000) and 50 (nucleotide positions 24,500-25,000) are provided in white text in each histogram.

At the sub-consensus level, 254 nonsynonymous intra-host single nucleotide variants (iSNVs) passing filtering criteria appeared in a single persistent infection while 15 nonsynonymous iSNVs reoccurred at a frequency of two persistent infections (Fig 3B, S3 Table). We observed iSNVs recurring in two separate infections at 15 different positions. One substitution of note was ORF1a:T1638I, which appeared as an iSNV in Case 29 and Case 41, eventually achieving consensus in Case 29. This ORF1a substitution was also separately observed as a consensus mutation in Case 48, which indicates that it was a convergent mutation at both the intra-host and consensus levels. Additionally, we found that the S2 subunit of the Spike gene showed an increased number of mutations at the sub-consensus level in Bin 50 (nucleotide positions 24,500–25,000), which was significantly enriched (14 iSNVs, adjusted $p = 0.042$) with nonsynonymous iSNVs (Fig 3B).

### Persistent BA.5.11 infection reveals intra-host viral subpopulations

One persistent infection (Case 60) yielded six sequences of the BA.5.11 (Omicron) lineage over a 103-day time span in 2023. Five sequences originated from anterior nares swabs, while the specimen source for one sequence collected on 2023-05-11 was unknown. These six consensus genomes demonstrated considerable divergence from the most closely related genomes in public repositories, which have collection dates about five months prior to the first sequence from this persistent infection case (Fig 4A). Interestingly, the sequences with collection dates of 2023-06-23 and 2023-07-11 appear nearly identical on the tree, while the sequences with collection dates of 2023-05-22 and 2023-07-08 also appear identical to one another but are clearly distinct from the other two. Allele frequencies of the sequences from this persistent infection case reveal an unusual pattern, with the same mutations appearing at similar frequencies between those collected on 2023-06-23 and 2023-07-11, distinct from the sites with similar frequencies between those collected on 2023-05-22 and 2023-07-08 (Fig 4B). This oscillating pattern of mutation and reversion, in combination with the phasing of alternate alleles in the aligned reads suggests the presence of at least two competing intra-host viral subpopulations.

### Persistent Epsilon infection demonstrates divergence from common ancestor

Two SARS-CoV-2 sequences from the same case (Case 33) originated from specimens collected more than a year apart (2021-04-29 and 2022-06-03). These genomes belonged to the B.1.429 (Epsilon) lineage, which is of particular interest considering that this lineage had disappeared from the United States as of August 2021, 10 months prior to the collection date of the second sequence. The specimen type differed between the two samples, as the first was a nasal swab and the second was a nasopharyngeal swab. The first sequence had 11 of 58 (19.0%) total variant sites with allele frequencies between 10–60%, and the second sequence had 47 of 143 (32.9%), but none of these mixed sites were shared between the sequences. Both consensus genomes appear to diverge from a common ancestor on a phylogenetic tree (Fig 5). There are seven amino acid substitutions in the first genome that are absent from the later genome, including two in the Spike protein (S:S255F, S:G339S), while there are 32 amino acid substitutions in the second genome absent from the first, with 10 in the Spike protein (S2 Table). The lack of shared iSNVs and shared consensus mutations between the two Epsilon genomes from the same persistent infection case suggest the emergence of two intra-host lineages, which likely diverged from a common ancestor for which no sequence data exists (Fig 5).

## Discussion

This study provides a framework for detection and characterization of persistent infections using routine genomic surveillance and clinical reporting data, while highlighting ongoing challenges to the comprehensive characterization of these unique cases. Using this framework, we identified 69 persistent infections among almost one million SARS-CoV-2 sequences from California. The methods and results described here can be used to enhance surveillance of persistent infections moving forward, which is of public health importance considering the impact previous highly divergent SARS-CoV-2 variants have had on the morbidity and mortality of COVID-19 worldwide.

PLOS Pathogens

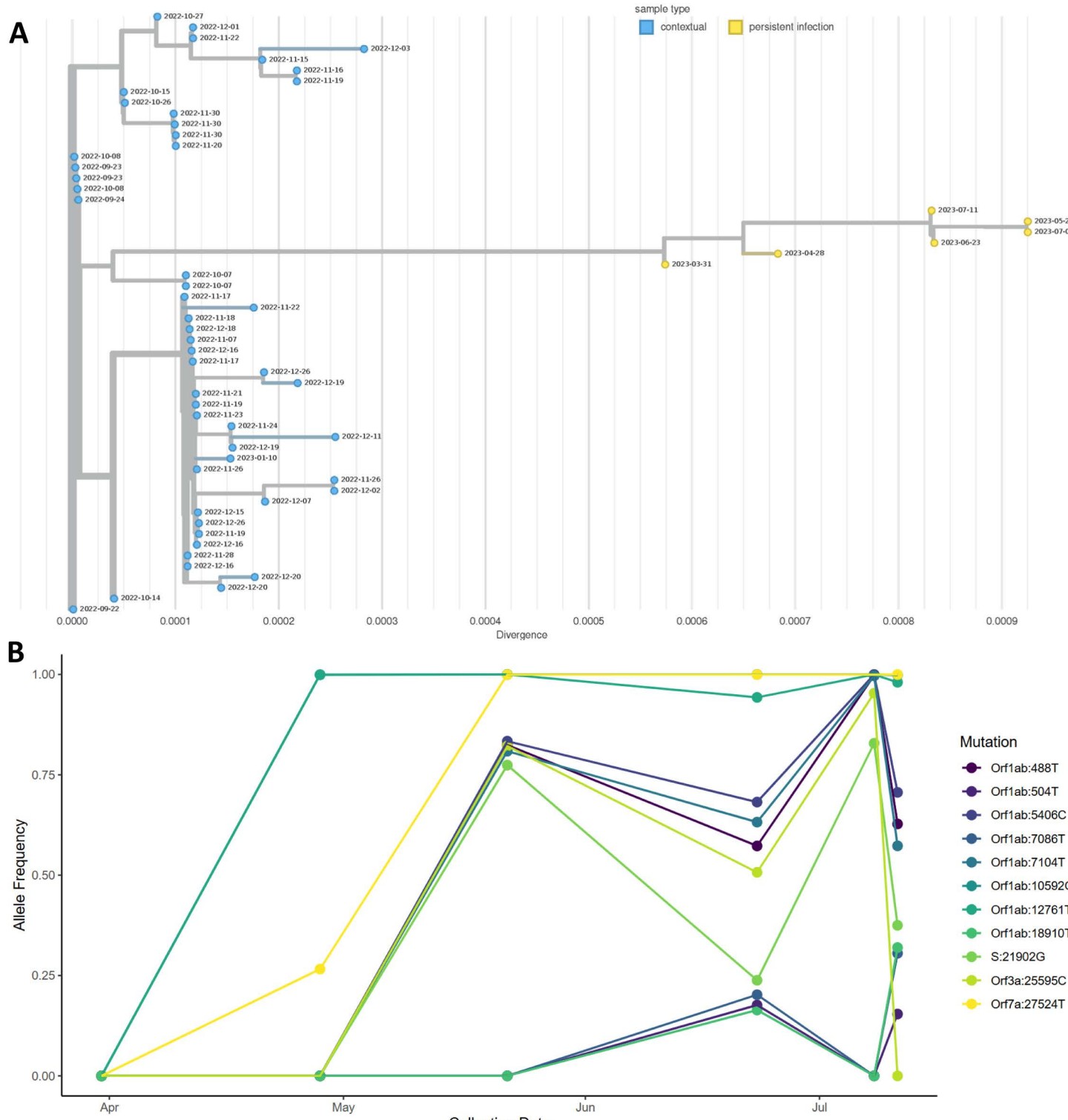

**Fig 4. Genomic diversity of a BA.5.11 persistent infection. A.** Phylogenetic tree of Case 60, a BA.5.11 persistent infection, including 50 contextual genomes from NCBI. The x-axis represents substitutions per site in the SARS-CoV-2 genome. Yellow nodes represent genomes from likely persistent infection and blue nodes represent contextual genomes from NCBI. The dates at the nodes indicate specimen collection date. **B.** Allele frequency plot showing changes in allele frequencies in the same host over time, with different colors representing different variants.

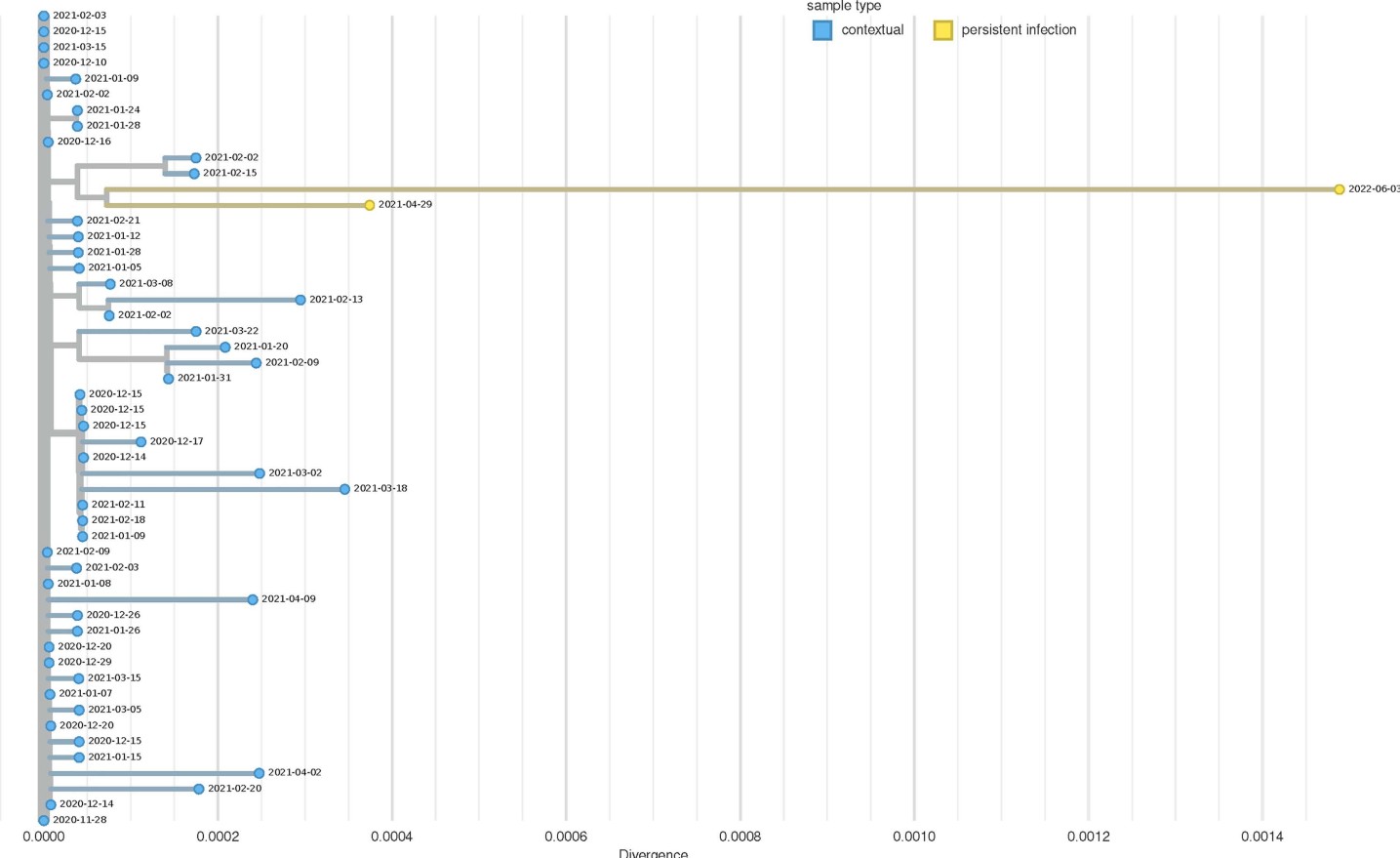

**Fig 5. Phylogenetic tree of a B.1.429 persistent infection with 50 contextual genomes from NCBI.** The x-axis represents substitutions per site in the SARS-CoV-2 genome. Yellow nodes represent genomes from likely persistent infection and blue nodes represent contextual genomes from NCBI.

There was a significant difference in the distribution of male and female persistent infection cases and all sequenced specimens ($x^2 = 4.5$, df = 1) (Table 1). This is consistent with previous findings in SARS-CoV-2 persistent infection cohorts [7]. The age distribution also differed significantly from that of all sequenced specimens ($x^2 = 81.401$, df = 3), where persistent infections were found more often among older patients. The subset of data with medical records showed that the persistent infection cases had underlying health conditions. This is not unexpected given similar findings from other studies [6,25,26]. Underlying health conditions may also explain the significant differences seen in the number of hospitalizations ($x^2 = 230.4$, df = 1) and deaths ($x^2 = 134.48$, df = 1) between persistent infection cases and all sequenced specimens in California. Further studies are needed to quantify demographic risk factors for persistent SARS-CoV-2 infections.

While the majority of persistent infection cases lasted less than a month (Fig 2A) and exhibited few genomic changes, several cases demonstrated considerable intra-host evolution (e.g., Figs 4 and 5, case 41, case 49 in S1 Table). The genomic characteristics of these cases were not dissimilar to other persistent infections reported in the literature, which are characterized by long branches on phylogenetic trees and accumulation of mutations over time [8,27]. None of these highly divergent persistent infection cases are known to have resulted in onward community transmission, which may be due to differences in viral fitness requirements for successful spread within a host versus between hosts [6].

Overall, we observed examples of convergent evolution and nonsynonymous mutation enrichment at the consensus level during persistent infections. Many of these substitutions were lineage-defining for Variants of Concern (VOCs) and

Variants of Interest (VOIs) that were either at one point dominant variants in the US [28,29], have been shown to confer host immune escape [30–33], or have been observed previously in persistent infections or saltation lineages [6,31] (Table 2). We observed both recurrent consensus mutations and iSNVs that have been previously documented in persistent infections. Notably, S:E484 and ORF1a:T1638I were convergent sites in this study, and these mutations have been observed repeatedly in past studies [6,7,31,34,35].

Consistent with previous reports, Spike RBD mutations, and in particular, substitutions at positions 444 and 484, appeared to be evolutionarily advantageous and were repeatedly observed in this study in apparent convergent evolution [30,36]. The substitutions we observed at these positions have been associated with immune evasion [30–33], including S:E484K, which has repeatedly appeared as a lineage-defining mutation in major lineages such as JN.1 and the Beta and Gamma VOCs [28]. The amino acid residue 444 in the Spike gene is notable for S:K444T, which was a lineage-defining mutation for the BQ.1 lineage that became dominant in the US in 2023 [28]. Interestingly, we only observed consensus mutations accumulating at position 444 in a BQ.1 lineage or in lineages circulating in the six months prior to BQ.1 becoming dominant (Table 2). The selective advantage seen in the BQ.1 lineage has been attributed to the S:K444T and S:N460K mutations [33,37]. Previous work has suggested that convergent mutations acquired over the course of persistent infections can indicate the evolutionary trajectory of future variants [7,38]. Considering this, the repeated substitutions we detected at S:444 are suggestive of a selective advantage in the BQ.1 lineage.

We observed somewhat different selective pressures at the sub-consensus level. Bin 50 (nucleotide positions 24,500–25,000) of the S2 subunit of the Spike gene was enriched for nonsynonymous iSNVs (Fig 3B). Previous reports have shown that iSNVs are more likely to be nonsynonymous than consensus mutations and more likely to be transient due to being deleterious [39,40]. This may explain why Bin 50 is not enriched for consensus mutations (Fig 3A) and why the convergent iSNVs we observed were all low prevalence mutations (<10% of sequences) among publicly available consensus genomes in GISAID [28]. It is also possible that host-specific factors contribute to these differences. In fact, it has been suggested that the recurrent ORF1a:T1638I mutation may be advantageous in specific within-host contexts but may be deleterious in other hosts or in terms of transmissibility [7,34].

We identified multiple persistent infection cases that did not follow a conventional pattern of ongoing evolution, in that later sequences were not direct descendants of the former. The phylogenetic tree of consensus assemblies from Case 60, the BA.5.11 persistent infection (Fig 4A), shows an oscillating pattern of reversions and mutations. This was further demonstrated through the analysis of allele frequencies, which was suggestive of competing subpopulations (Fig 4B). Direct evidence of phasing accords with the hypothesis of competing sublineages. For example, nucleotide positions 7086 and 7104 in the 2025-07-11 sample from Case 60 have complementary mutations; one set of sequencing reads has the Wuhan-1 genome's nucleotide at 7086 and a variant at 7104, while another set shows the converse. A previous study identified a similar pattern of mutations fixing and reverting in a persistent infection immediately following treatment with remdesivir and convalescent plasma [41]. Unfortunately, treatment information was not available for this case, but it seems possible that selective pressures outside of the host immune system may also be contributing to intra-host evolution during persistent infections through changes in fitness requirements of viral subpopulations.

In Case 33, the B.1.429 persistent infection, the two genomes seemingly diverged from a common ancestor within the host (Fig 5). Differential tissue tropism might be contributing to the level of divergence between the sequences in this case, since the two sequences originated from different specimen types. Previous studies have identified differences in SARS-CoV-2 infectivity between olfactory and respiratory epithelial cells, and between cell types expressing different receptors such as ACE2 or TMPRSS2, with particular variants of SARS-CoV-2 [8,42,43]. Both Case 33 and Case 60 were likely infected for a period of time before the collection dates of the first sequences, based on the divergence observed from the most closely related genomes in public repositories to the first consensus genome (Figs 4A and 5). These evolutionary patterns and complex subpopulation structures further demonstrate the value of using both epidemiologic and genomic criteria when identifying potential persistent infection cases.

One of the main limitations of this study is that we were unable to establish completely systematic criteria for identifying persistent infections. We attempted to set specific quality control thresholds for genome assemblies as well as divergence thresholds for genome assemblies within a single case but found that those criteria resulted in the inadvertent exclusion of many cases we believed to be persistent infections. For example, we frequently saw a high number of ambiguous sites within genome assemblies of persistent infections. Sometimes these ambiguous sites were due to mixed alleles, which are common in persistent infections due to intra-host evolution, but in other cases they were due to legitimate sequence quality issues such as amplicon dropout. We also aimed to identify as many persistent infections as possible, in order to gain a better understanding of epidemiologic characteristics associated with persistent infections. Therefore, an entirely systematic process was not necessary for this study, but future efforts could be directed towards developing a systematic process to assist with ongoing surveillance of persistent infections. With respect to accumulated mutations and iSNV analyses, although every mutation included was manually reviewed, it is possible that some false positives caused by sequencing artifacts were not filtered out; we do not believe, however, that such artifacts contributed to the findings in this study.

There are also some limitations relating to the epidemiologic data used in this study. First, the IGED is derived from ELRs, which are subject to several known challenges including data quality concerns and inconsistency across data sources [44]. In particular, inconsistencies in the input fields used in the Epi Criteria (Fig 1) may have resulted in the inadvertent exclusion of some persistent infection cases. Nonetheless, we identified a significant number of cases we were confident were persistent infections using the IGED, 69 of which were later confirmed using genomic analyses. Second, hospitalizations and deaths due to COVID-19 rather than from other causes while infected with COVID-19 were not differentiated. Third, we were unable to distinguish between unvaccinated cases and cases with unknown vaccination status. Hospitalizations and deaths may be underreported from cases occurring before California's enactment of a law requiring hospitals to report these cases on July 27, 2021. Lastly, we were prohibited from linking epidemiologic information (age, sex, pertinent underlying health conditions, etc.) to sequence accession numbers in public repositories, to preserve patient privacy and confidentiality.

Overall, the number of persistent infections identified in this study is most likely an underestimation of the true number of persistent infections in California for several reasons: 1) not every case of SARS-CoV-2 is sequenced, therefore we were limited to genomes that were available from public health laboratories in California and public data sources; 2) only a portion of genomes had fastq files available; 3) at least two sequences from the same case were required; and 4) putative persistent infection cases that had negative PCR tests in between PCR-positive collection dates were excluded, although some studies have shown that rebound of the original infection can occur after negative tests [6,45].

Here, we have characterized SARS-CoV-2 persistent infections in California, described intra-host evolutionary dynamics, and provided genomic and epidemiologic insights that are often missed when analyses are focused on individual cases. The identification of persistent infections will remain important with the decline of routine sequencing of SARS-CoV-2, with smaller sample sizes limiting our ability to detect and monitor emerging variants. The findings from this study can be used to enhance existing surveillance systems, direct sequencing efforts, and expedite the analysis of putative persistent infections in the hopes of assessing possible public health impact before highly divergent variants become globally disseminated.

## Methods

### Ethics statement

The genomic surveillance work of California COVIDNet was reviewed by the State of California—Health and Human Services Agency Committee for the Protection of Human Subjects and exempted under project number 2023–103. The Research Determination Committee for the Kaiser Permanente Northern California region determined the project did not meet the regulatory definition of research involving human subjects per 45 CFR 46.102(d).

## Epidemiologic and health data sources

To match SARS-CoV-2 sequences to epidemiologic information, data were obtained from four different CDPH databases: the COVID-19 Hospitalization Registry, the COVID-19 Case Registry, the Vaccine Registry, and the IGED. The IGED is a database combining California SARS-CoV-2 lineages derived from WGS with patient demographic and epidemiologic information from the COVID-19 Case Registry [46]. The COVID-19 Hospitalization Registry includes data reported to CDPH following an All Facilities Letter that required all hospitals in California to report specific patient-level information for each hospitalized patient who tested positive for COVID-19 effective July 27, 2021 or later. Therefore, any cases with an unknown hospitalization status in CDPH databases were considered not hospitalized, and any cases with unknown death status were considered not to have died for the purposes of this study. The COVID-19 Case Registry includes SARS-CoV-2 laboratory results that are reported electronically or manually by laboratories, healthcare providers, and local health departments. The Vaccine Registry database includes vaccine information reported to the California Immunization Registry. A case was considered vaccinated if a dose of COVID-19 vaccine was received 14 days or more before the earliest known positive specimen collection date. Unknown vaccination status and unvaccinated status were not distinguishable and thus not included.

Separately, patient records in the California COVIDNet patient databases were queried for medical record numbers. These were collated and integrated with the cases found via querying the IGED. Available medical records from Kaiser Permanente Northern California [47,48] were reviewed by a non-physician data scientist to determine whether the case may have had a compromised immune system and if antiviral drugs were prescribed. The medical records were purposefully not linked to sequence accession numbers in this study in deference to confidentiality concerns and privacy regulations. Demographic and epidemiologic data were compared between persistent infection cases and all sequenced SARS-CoV-2 cases from California. For the dataset of all sequenced SARS-CoV-2 cases from California, specimen collection month was derived from the earliest positive specimen collection date in the IGED. Total case counts were taken from the California COVID-19 Case Registry and only included confirmed cases with available sequence data. A confirmed case was defined as an individual with detection of SARS-CoV-2 ribonucleic acid (RNA) in a clinical or post-mortem specimen using a diagnostic molecular amplification test performed by a Clinical Laboratory Improvement Amendments (CLIA)-certified provider, or detection of SARS-CoV-2 RNA in a clinical or post-mortem specimen by genomic sequencing.

To compare the epidemiologic data distribution of the persistent infection cases with all sequenced SARS-CoV-2 cases in California, Pearson's Chi-squared test with Yates' continuity correction was performed using the chisq.test function in RStudio v4.4.1 [49]. For the age distribution chi-square test, the ≤ 17 age category was not included due to low sample size.

## Identification of putative persistent infections using epidemiologic case data

We defined the time period of a persistent infection as a minimum of 21 days, indicated by successful sequencing of specimens with collection dates at least 21 days apart. Because our determinations are based on successfully sequenced and assembled SARS-Cov-2 samples, 21 days is the lower limit for infection period in each case, with onset dates presumably occurring no later than the earliest sample, and end of infection occurring sometime later than the last sample. We used a 21 day threshold as studies have shown that viral shedding and culturable virus are undetectable by then in the vast majority of patients with normal acute SARS-CoV-2 infections [50–52] and 21 days is conveniently exact in terms of weeks.

A list of specimens was extracted from the IGED according to these filtering criteria: 1) cases had multiple identifiable SARS-CoV-2 sequences associated with them; 2) cases passed metadata quality filters by having complete specimen collection dates, lineages, and sequence identifiers in the IGED; 3) sequences from those cases had the same WHO variant designation (Alpha, Omicron, etc.) and NextClade clade; and 4) the time difference in collection dates between first and last sequences was ≥ 21 days (Fig 1). Finally, patients were excluded if any negative tests were identified between their sequences, suggesting the patient had cleared the initial infection and was later reinfected.

## Genome assembly and characterization

Raw Illumina (San Diego, CA, USA) and Element Biosciences (San Diego, CA, USA) AVITI sequence data generated by CDPH were analyzed using the TheiaCoV Workflows for Genomic Characterization (https://github.com/theiagen/public_health_bioinformatics) that corresponded to the library layout, paired-end or single-end [53,54]. Default options were used, with the exception of a minimum depth requirement of 20x (min_depth = 20), minimum frequency for including a variant site in the iVar report of 0.01 (variant_min_freq = 0.01), and adjustments to the minimum read length based on the length of amplicons used to generate the data (trim_minlen) [55]. The sequencing platform, amplicon scheme and optional parameters used for all CDPH sequences are included in S1 Table. Links to associated protocols for the different amplicon schemes are included in S4 Table.

Sequence data generated by CDPH on the Clear Labs platform were assembled using the default, proprietary pipeline within the Clear Labs portal. This pipeline was selected as it contained automated correction for insertion and deletion events commonly seen in Clear Labs generated sequence data. Post-assembly characterization was performed using the TheiaCoV_FASTA workflow with default settings. The corresponding amplicon scheme is also included in S1 Table.

For sequence data from California generated outside of CDPH, either by local public health laboratories, laboratories contracted by the US Centers for Disease Control and Prevention, or private entities, raw reads were re-processed using v1.3.0 of the TheiaCoV workflow that corresponded to the sequencing platform, using the same options mentioned above. One exception to this method was made for LabCorp-generated data which was sequenced using a PacBio instrument for which a TheiaCoV workflow does not exist. These sequence data were not reassembled, but the raw reads were still available for investigation into allele frequencies. Since all sequence data generated outside of CDPH, aside from that one exception, were reanalyzed, the resulting genome assemblies used for this study may have differed from those available in public repositories which were uploaded by the original sequencing laboratories.

## Genomic classification of cases as persistent infections or reinfections

To classify incidents as persistent infections or reinfections, we evaluated each incident according to four key considerations: divergence between the genome assemblies, divergence between the genome assemblies and those in public repositories, genome assembly quality, and changes in allele frequencies. These evaluations involved placing the genome assemblies on global trees as well as subtrees with 5,000 of the most closely related sequences in GISAID, GenBank, COG-UK, and CNCB using the UShER web tool (https://genome.ucsc.edu/cgi-bin/hgPhyloPlace). On the global tree, if the earlier genome assembly appeared ancestral to the later assembly (or assemblies), persistent infection was most likely, but if the assemblies instead shared a common ancestor that was more than a few mutations divergent, reinfection was more likely. When assessing subtrees, if the later assembly was identical or within a few mutations of divergence to global contextual assemblies, reinfection was most likely unless the time window between collection dates was short enough where divergence would not be expected, based on an approximate substitution rate of 2–3 nucleotides/month extrapolated from prior studies [22,56]. However, there are limitations to phylogenetic placement when genome assemblies have a large number of Ns, so genome assemblies from the same putative persistent infection were also visualized using the Nextclade v3.8.2 web tool (https://clades.nextstrain.org/) to assess the presence and absence of certain mutations if needed. Specific genome assembly quality thresholds were not used, and for cases where genome assembly quality came into question, further investigation using read data was performed.

Read data were investigated to identify the reason for Ns in the genome assemblies and determine whether that reason justified exclusion from the study. For genome assemblies that had strings of Ns spanning hundreds of nucleotides, low sequencing depth (potentially due to amplicon dropout) was the likely reason for Ns, and this was confirmed by visualizing the binary alignment map (BAM) file output from the TheiaCoV workflow in IGV within Terra.bio relative to the Wuhan-1 reference genome (MN908947). However, low sequencing depth in specific regions alone was not justification for exclusion in all cases, particularly if the high-quality regions of the genome were nearly identical between sequences

from the same case. Additionally, we conducted further investigation into genomic sites with mixed alleles from the iVar TSV output. If there was evidence of single nucleotide variants (SNVs) gradually arising to fixation demonstrated by an increase in alternative allele frequency, or reversion demonstrated by a decrease in alternative allele frequency, we considered those as evidence in favor of a persistent infection. Additionally, if there were minor alleles shared between sequences of the same case, we also considered that to be evidence in favor of a persistent infection. However, we were careful to distinguish mixed sites due to intra-host evolution from potential contamination by looking for phasing of SNVs in the reads aligned to Wuhan-1 in IGV within Terra.bio. If we were unable to confidently determine whether a case was a persistent infection or reinfection based on the previous considerations, the case was excluded. Importantly, these genomic considerations were not without subjectivity and relied on subject-matter expertise of bioinformaticians familiar with SARS-CoV-2 data, but supplementary materials have been provided as guidance for those looking to perform similar analyses (S1 File).

To visualize aggregated collection date and clade information from all persistent infection cases in this study, a dot plot was created using the tidyverse v2.0.0 and svglite v2.2.1 packages in RStudio v4.3.3 (Fig 2A) [49,57,58].

## Mutation data processing

To calculate consensus mutation statistics for persistent infections and reinfections, a list of nucleotide substitution, insertion, and deletion mutations was parsed from NextClade v2.14.0 (dataset 2024-01-15T12:00:00Z) results for all available consensus assemblies [59]. In order to focus on functional changes in the genome, only nonsynonymous amino acid substitutions were considered for mutation analyses. All sequences were included for consensus mutation analyses regardless of the instrument or technology used to perform sequencing. For iSNV analyses, only Illumina and Element Biosciences sequences were considered due to the lower read depth and higher error rate associated with long-read sequencing data.

## Masking problematic mutations

In order to minimize the inclusion of erroneous variant calls due to sequencing or bioinformatic errors, we masked specific consensus and iSNV mutations. In particular, each sequence was assigned a lineage by NextClade, and any lineage-defining mutations or fixed mutations within the lineage that were observed as accumulated consensus mutations or iSNVs were excluded. Mutations associated with mismapping of reads at insertion and deletion sites were also excluded. Additionally, nonsense consensus mutations outside of ORF8 were excluded because they were likely sequencing arti-facts. Nonsense mutations in ORF8 were retained since it is known to be a region prone to truncation and gene knockout [60]. We also masked any mutations that have been previously marked as problematic (https://github.com/W-L/Problemat-icSites_SARS-CoV2/blob/master/problematic_sites_sarsCov2.vcf). Finally, all mutations were also subject to manual review in an attempt to further reduce potential bias from erroneous mutations in primer binding regions or loci covered by the terminal ends of reads, and those excluded are listed in S5 Table.

## Identifying recurrent accumulated consensus mutations

To identify mutations that repeatedly arose during persistent infections, we counted amino acid substitutions that appeared in the second or later consensus genome but not in the initial genome from each case. These mutations were considered "accumulated" during a persistent infection. These mutations were counted once per position per case in order to account for the variable number of sequences per case. We therefore counted the number of infections in which non-synonymous mutations occurred at each genomic location. In addition, we excluded mutation counts at any positions where the initial consensus genome in a case had a missing or ambiguous base call. A complete list of these mutations as well as the Pango lineage called by NextClade for each consensus sequence is provided in S2 Table.

## Identifying recurrent iSNVs

We selected nonsynonymous iSNVs between 5% and 60% (sub-consensus) allele frequency with a minimum alternative allele coverage depth of 20x and a minimum total depth of 100x. In order to minimize potential bias due to sequencing artifacts, iSNVs were excluded if the alternative alleles had a greater than 10-fold strand bias, or if the reference alleles had a greater than 10-fold strand bias and reference read depth greater than 20. We assumed that each persistent infection would start with few or no iSNVs because SARS-CoV-2 has been shown to have low intra-host diversity upon initial infection due to a tight transmission bottleneck [61]. As such, iSNVs present in any of the persistent infection genomes from a single case, including the first genome, were treated as having "accumulated" since initial infection. For Fig 3B, as with the consensus-level analysis, the genomic position of each amino acid change is counted only once per patient in order to account for the variable number of sequences per case. In effect, this measured the number of infections in which non-synonymous iSNVs were identified at each genomic location. A complete list of these iSNVs as well as the Pango lineage called by NextClade for each consensus sequence is provided in S3 Table.

## Binned mutation distributions

To evaluate whether certain regions of the SARS-CoV-2 genome accumulated more mutations than others, we divided the genome into 500 nucleotide bins as previously described [6,7]. Each bin includes all amino acid substitutions for which the position of the first nucleotide of the codon is within that bin. In other words, a bin denoted as $bin_i$ includes all mutations in nucleotide positions $[i*500, i*500+500)$. We used a one-tailed binomial test from the MutationalPatterns package v3.14.0 with Bonferroni multiple test correction to determine which bins were significantly enriched for nonsynonymous mutations (adjusted $p < 0.05$) [62]. Lollipop plots and histograms were generated in R version 4.4.1 with ggplot2 v3.5.1 [49,63].

## Phylogenetic trees and allele frequency plot

A concatenated fasta file containing all persistent infection genomes was uploaded to UShER (https://genome.ucsc.edu/cgi-bin/hgPhyloPlace) [64] to demonstrate placement on the global SARS-CoV-2 phylogenetic tree (Fig 2B). Phylogenetic trees used to visualize individual persistent infection cases were constructed using all SARS-CoV-2 genomes collected from the same persistent infection, combined with 50 contextual genomes from GenBank, using the Augur_PHB workflow within the Public Health Bioinformatics repository (https://github.com/theiagen/public_health_bioinformatics) (Figs 4A and 5). The contextual genomes were selected for each persistent infection case by placing the genome with the earliest collection date on an UShER subtree containing 50 genomes from GenBank, COG-UK, and CNCB (https://genome.ucsc.edu/cgi-bin/hgPhyloPlace)(accessed March 8, 2024). The contextual genome with the earliest collection date was selected as the reference for building the final phylogenetic tree for each persistent infection set. The resulting phylogenetic trees were visualized in Auspice, annotated by location and collection date (https://auspice.us/). In order to create the figures, the trees were downloaded as SVG files from auspice, and the SVG files for each were directly edited in order to move the legend to the right so that the top edges of the trees were not occluded. No manipulations of the trees themselves were made.

To generate an allele frequency plot for Case 60, the variant calling results for all sequences from that case were combined. Masking was performed to remove any mutation with depth of coverage < 20x, putative substitutions immediately adjacent to insertions or deletions, and mutations with allele frequencies <10%. Mutations with a cumulative frequency of <25% summed across all sequences from the case were also excluded, as well as those at >95% frequency in every sample. Remaining mutations were visualized by their allele frequencies over time using R Version 4.4.1 with ggplot2 v3.5.1 [49,63].

## Supporting information

**S1 File. Guide to aid classifying persistent infection using genomic considerations.**
(DOCX)

**S1 Fig. Map of California persistent SARS-CoV-2 infections per Public Health Officer (PHO) Regions.** N is the count of persistent SARS-CoV-2 infections identified in this paper. PHO regions are colored according to 2023 population size (worldpopulationreview.com accessed 4 March 2024). Map was created using ArcGIS Pro (version 3.0.0). The original County lines were downloaded from https://catalog.data.gov/dataset/ca-geographic-boundaries. This is an open access dataset in the public domain and no license information is provided.
(PNG)

**S1 Table. List of sequencing data of the persistent infection cases.** This includes Sample IDs, Case IDs, GISAID accessions, and other relevant information. Each Case ID is one persistent infection.
(XLSX)

**S2 Table. List of 201 accumulated mutations identified in the 69 persistent infection cases.**
(CSV)

**S3 Table. List of 284 intra-host single nucleotide variants (iSNVs) identified in the 69 persistent infection cases.**
(CSV)

**S4 Table. List of the different amplicon schemes used in the persistent infection dataset and the links to the associated protocols.**
(XLSX)

**S5 Table. List of 9 mutations removed after manual review.**
(CSV)

## Acknowledgments

We gratefully acknowledge all data contributors, i.e., the Authors and their Originating Laboratories responsible for obtaining the specimens, and their Submitting Laboratories for generating the genetic sequence data and metadata and sharing via the GISAID Initiative, on which this study is based. We would like to thank the local California Association of Public Health Laboratory Directors, public health laboratorians, and California COVIDNet lab partners for their contributions to California COVIDNet. We thank the CDPH CalREDIE team, the CDPH COVIDNet laboratory group, CDPH Data team, the CDPH COVID Clinical team and CDPH Epidemiologists for their involvement in data entry, data definitions, database creation, and valuable feedback on this work. Thank you to Esther Lim for her valuable help and feedback. Lastly, we thank the team of international volunteers that propose and designate new Pango lineages.

## Author contributions

**Conceptualization:** John M. Bell, Jesse Elder, Rahil Ryder, Emily A. Smith.

**Data curation:** John M. Bell, Jesse Elder, Rahil Ryder, Emily A. Smith.

**Formal analysis:** John M. Bell, Jesse Elder, Rahil Ryder, Emily A. Smith, Michelle Scribner.

**Funding acquisition:** Debra A. Wadford.

**Investigation:** John M. Bell, Jesse Elder, Rahil Ryder, Emily A. Smith.

**Methodology:** John M. Bell, Emily A. Smith.

**Project administration:** Debra A. Wadford.

**Resources:** Sabrina Gilliam, Deva Borthwick, Megan Crumpler, Jacek Skarbinski, Christina Morales, Debra A. Wadford.

**Supervision:** Christina Morales, Debra A. Wadford.

**Validation:** John M. Bell, Jesse Elder, Rahil Ryder.

**Visualization:** Jesse Elder, Rahil Ryder, Emily A. Smith, Michelle Scribner.

**Writing – original draft:** John M. Bell, Jesse Elder, Rahil Ryder, Emily A. Smith.

**Writing – review & editing:** John M. Bell, Jesse Elder, Rahil Ryder, Emily A. Smith, Debra A. Wadford.

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
