## [Decision Letter · Decision Letter 0]

22 Aug 2025

Genomic and epidemiologic characteristics of SARS-CoV-2 persistent infections in California, January 2021 - July 2023

PLOS Pathogens

Dear Dr. Wadford,

Thank you for submitting your manuscript to PLOS Pathogens. After careful consideration, we feel that it has merit but does not fully meet PLOS Pathogens's publication criteria as it currently stands. Therefore, we invite you to submit a revised version of the manuscript that addresses the points raised during the review process.

Please submit your revised manuscript within 60 days Oct 21 2025 11:59PM. If you will need more time than this to complete your revisions, please reply to this message or contact the journal office at plospathogens@plos.org. Please include the following items when submitting your revised manuscript:

We look forward to receiving your revised manuscript.

Kind regards,

Denis Kainov

Academic Editor

PLOS Pathogens

Alexander Gorbalenya

Section Editor

PLOS Pathogens

Editor-in-Chief

PLOS Pathogens

orcid.org/0000-0003-2946-9497

Editor-in-Chief

PLOS Pathogens

orcid.org/0000-0002-7699-2064

**Additional Editor Comments:**

I’m editing your manuscript PPATHOGENS-D-25-01692, "Genomic and epidemiologic characteristics of SARS-CoV-2 persistent infections in California, January 2021 - July 2023," and recommended "Major Revision".

Reviewer 1 recommends "Accept," praising the manuscript’s clarity. Reviewer 3 suggests revisions to define "persistent infection" and improve data presentation. Reviewer 2 recommends "Reject," citing issues with the 21-day criterion for persistent infection and with positive selection analysis. I give opportunity to the authors to address these concerns through revisions, including justifying or redefining the persistent infection criterion, refining the selection analysis, and enhancing clarity.

**Journal Requirements:**

**Reviewers' Comments:**

Reviewer's Responses to Questions

**Part I - Summary**

Reviewer #1: The authors presented the epidemiological and genomics characteristics of persistent infections in COVID-19 patients. The manuscript is well written and described. The presence of various mutations in the persistent infections are discussed thoroughly.

Reviewer #2: The authors characterized persistent infection cases in California using routine epidemiologic and genomic surveillance data. A total of 69 persistent infection cases, ranging from 21 to 400 days with an average of 44 days in duration, were detected. Significant differences in age distribution, sex, hospitalizations, and death between persistent infection cases and all cases. The author claimed that genomic data suggested positive selection in the Spike receptor binding domain and convergent evolution toward immune evasion.

Reviewer #3: This paper studied the persistent infection through genomic data monitoring and analyzed of 69 cases of SARS-CoV-2 persistent infection collected during the pandemic from January 2021 to July 2023. Overall, research has benefited other public health organizations in studying the highly differentiated SARS-CoV-2 virus.

However, several areas require attention to ensure the manuscript meets the standards for publication.

**Part II – Major Issues: Key Experiments Required for Acceptance**

Reviewer #1: Nil.

Reviewer #2: Analyses done in this study have a critical problem due to the criteria for detecting persistent SARS-CoV-2 infections.

A 2024 paper by Machkovech et al, which is the reference 1 in the manuscript, defined the persistent SARS-CoV-2 infection as follows: At least one of the following for at least 30 days after symptom onset: 1) isolation of replication-competent virus, 2) accumulation of viral genetic changes over time, 3) cycle threshold values 30 or lower, for 30 days or longer or antigen test positivity, 4) detectable subgenomic RNA, 5) ongoing symptoms.

However, the selection criteria for persistent SARS-CoV-2 shown in Figure 1 in this manuscript are looser than Machkovech’s definition. Even if two sequences were isolated from a patient with more than 21 days' difference, this did not necessarily mean the second sequences were collected at least 30 days after symptom onset. The authors should have described the reasons why they set the criteria for detecting persistent SARS-CoV-2 infections as shown in Figure 1.

The characteristics of the SARS-CoV-2 infections described in this manuscript may not be those of persistent SARS-CoV-2 infections but those of patients from whom viral sequences were collected more than 21 days apart.

The reviewer is wondering why the authors did not detect persistent SARS-CoV-2 infections by comparing the date of symptom onset and the date of sample collection for sequencing. This will be a more straightforward criterion to detect persistent SARS-CoV-2 infections using epidemiological data and sequencing data than those used in this manuscript.

The analysis of positive selection in this manuscript also has a problem. Positively selected codons should be detected by comparing the non-synonymous and synonymous substitution rates. See, for instance, a research of influenza A viruses by Bush et al.

References

Machkovech HM, Hahn AM, Garonzik Wang J, Grubaugh ND, Halfmann PJ, Johnson MC, Lemieux JE, O'Connor DH, Piantadosi A, Wei W, Friedrich TC. Persistent SARS-CoV-2 infection: significance and implications. Lancet Infect Dis. 2024 Jul;24(7):e453-e462. doi: 10.1016/S1473-3099(23)00815-0. Epub 2024 Feb 7. PMID: 38340735.

Bush RM, Fitch WM, Bender CA, Cox NJ. Positive selection on the H3 hemagglutinin gene of human influenza virus A. Mol Biol Evol. 1999 Nov;16(11):1457-65. doi: 10.1093/oxfordjournals.molbev.a026057. PMID: 10555276.

Reviewer #3: The analysis of the results is currently a weak point of the article.

Authors are advised to provide a definition of the concept of "persistent infection" when it first appears.

**Part III – Minor Issues: Editorial and Data Presentation Modifications**

Reviewer #1: Nil.

Reviewer #2: The authors of this manuscript should have avoided using the terms "persistent infection" and "positive selection” because of the reasons described above.

Reviewer #3: Author Summary:

Line 63: A comma should be added after i.e., and the author is advised to modify it

Retrospective analysis revealed 69 persistent infection cases in California

Line 116: For smaller numbers, usually numbers less than 10, it is better to spell them in English, and they are also formal.

I recommend that authors review and standardize digital writing throughout the manuscript.

Medical records indicate presence of comorbidities in subset of persistent infection cases

Patients with cancer should be analyzed for co-infection in cases with persistent infection, whether the mutation of positive selection of immune escape is due to cancer, and whether the therapeutic agents used in each case have an effect on mutation accumulation.

I suggest that the author enhance the clarity of Figures to improve the visual quality of the presentation.

This modification will help improve the clarity and appeal of the graphical representation.

PLOS authors have the option to publish the peer review history of their article (what does this mean? ). If published, this will include your full peer review and any attached files.

**Do you want your identity to be public for this peer review?** For information about this choice, including consent withdrawal, please see our Privacy Policy .

Reviewer #1: No

Reviewer #2: No

Reviewer #3: No

**Figure resubmission:**

**Reproducibility:**



---

## [Editor Report · Decision Letter 1]

29 Oct 2025

Dear Debra,

We are pleased to inform you that your manuscript 'Genomic and epidemiologic characteristics of SARS-CoV-2 persistent infections in California, January 2021 - July 2023' has been provisionally accepted for publication in PLOS Pathogens.

Best regards,

Denis Kainov

Academic Editor

PLOS Pathogens

Alexander Gorbalenya

Section Editor

PLOS Pathogens

Sumita Bhaduri-McIntosh

Editor-in-Chief

PLOS Pathogens

orcid.org/0000-0003-2946-9497

Michael Malim

Editor-in-Chief

PLOS Pathogens

orcid.org/0000-0002-7699-2064

Editor's comments:

This study retrospectively analyzes California genomic and epidemiologic surveillance to identify and characterize persistent SARS-CoV-2 infections: 69 cases with 21–400 days between sampled specimens. It includes hospitalizations, deaths, and immunocompromising comorbidities. The authors demonstrate intrahost evolution with enrichment of mutations in the Spike RBD and convergent changes (notably at K444 and E484), and they present illustrative cases, including a 400-day Epsilon infection and a BA.5.11 infection showing competing intrahost subpopulations. The work could provide a framework for detection and monitoring of persistent infections and for prediction of virus evolution.

The authors have convincingly addressed the key reviewer comments: they justified the ≥21-day threshold and added clarifying text in the Introduction, Results, and Methods; they replaced some terms such as “positive selection” vs “mutation enrichment” throughout the manuscript and figure captions; they standardized style and improved figure clarity, removing a panel and adding explanations about phasing/reversions; and they reasonably declined an additional oncology-focused analysis as outside the scope of a surveillance study and constrained by privacy. But they acknowledged this limitation. With these revisions, the reviewers’ concerns are resolved, and the manuscript is suitable for acceptance in its current form.
---

## [Editor Report · Acceptance letter]

Dear Dr. Wadford,

We are delighted to inform you that your manuscript, "Genomic and epidemiologic characteristics of SARS-CoV-2 persistent infections in California, January 2021 - July 2023," has been formally accepted for publication in PLOS Pathogens.

Best regards,

Sumita Bhaduri-McIntosh

Editor-in-Chief

PLOS Pathogens

orcid.org/0000-0003-2946-9497

Michael Malim

Editor-in-Chief

PLOS Pathogens

orcid.org/0000-0002-7699-2064